# Impact of Polyvascular Disease on Long-Term Prognosis of Patients with Acute Coronary Syndrome—A Retrospective Cohort Study in Italy

**DOI:** 10.3390/jcm14124158

**Published:** 2025-06-11

**Authors:** Gian Francesco Mureddu, Stefano Rosato, Paola D’Errigo, Pompilio Faggiano, Paola Ciccarelli, Gabriella Badoni, Marco Ambrosetti, Francesco Fattirolli, Giovanni Baglio

**Affiliations:** 1Cardiac Rehabilitation Unit, San Giovanni-Addolorata Hospital, 00184 Rome, Italy; 2ITACARE-P (Italian Alliance for Cardiovascular Rehabilitation and Prevention), 21100 Varese, Italy; cardiologia@pompiliofaggiano.it (P.F.); francesco.fattirolli@unifi.it (F.F.); 3National Center for Global Health, Istituto Superiore di Sanità, 00161 Rome, Italy; stefano.rosato@iss.it (S.R.); paola.derrigo@iss.it (P.D.); gabriella.badoni@iss.it (G.B.); 4Cardiology Unit Poliambulanza Foundation, 25124 Brescia, Italy; 5Scientific Secretariat of the Presidency, Istituto Superiore di Sanità, 00161 Rome, Italy; paola.ciccarelli@iss.it; 6Cardiac Rehabilitation Unit, Crema Hospital, 26013 Cremona, Italy; 7Department of Clinical and Experimental Medicine, University of Firenze, 50134 Florence, Italy; 8Italian National Agency for Regional Healthcare Services, 00187 Rome, Italy; baglio@agenas.it

**Keywords:** polyvasculopathy, acute coronary syndrome, peripheral artery disease, cerebrovascular disease, epidemiology

## Abstract

**Background**: Atherothrombosis is a systemic disease that may affect one or more than one vascular bed. Data on the impact of polyvascular disease (PVD) on the long-term prognosis of patients with coronary artery disease (CAD) are still scarce. Aim: To assess the prevalence of symptomatic PVD in a cohort of patients with a new episode of acute coronary syndrome (ACS) and to investigate the impact of multiple vascular beds involvement on long-term outcomes. **Methods**: We analysed a nationwide, comprehensive administrative database of consecutive patients aged ≥ 40 years admitted for a new episode of ACS in Italy in 2017–2018. Patients with ACS were stratified according to the presence of peripheral artery disease (PAD) only; cerebrovascular disease (CeVD) only; PAD+CeVD; or neither (no PAD/noCeVD, i.e., ACS only). A multivariate Cox proportional hazards model was used to assess the impact of PAD only; CeVD only and PAD+CeVD on 5-year MACCE. **Results**: A total of 342,052 patients hospitalised with ACS were identified. Among them, 24,727 (7.2%) were patients with PAD only, 16,887 (4.9%) with CeVD only, and 5810 (1.7%) with PAD+CeVD. After adjusting for age, sex, and comorbidities, the hazard ratio (HR) for 5-year MACCE was 1.37 (95% CI: 1.35–1.40), 1.36 (95% CI: 1.33–1.39), and 1.45 (95% CI: 1.40–1.50) in patients with PAD only, CeVD only, and PAD+CeVD, respectively, compared with patients with ACS only. **Conclusions**: In patients with ACS, the involvement of a second vascular bed increases the risk of long-term outcomes; the simultaneous involvement of three vascular beds further increases the risk of long-term outcomes.

## 1. Introduction

Atherosclerotic cardiovascular disease (ASCVD) is a systemic condition that can affect the coronary, cerebral, and peripheral arteries. It represents the most common cause of death globally with an increasing prevalence and incidence worldwide [1]. It may manifest subclinically for many years before leading to an acute event at a single vascular site. The global prevalence of subclinical cerebrovascular disease is estimated to be around 20% in the general population, equivalent to over 800 million people, whereas the prevalence of symptomatic cerebrovascular vascular disease (CeVD) is estimated to be around 1.5% [2]. The prevalence of peripheral arterial disease (PAD) in people aged > 25 years is reported to be around 5.5% and increases exponentially with age, affecting approximately 200 million people worldwide [3]. Patients with polyvascular disease (PVD), a condition affecting more than one vascular bed, show a high risk of major adverse cardiovascular events (MACCE) such as myocardial infarction, stroke, and cardiac death [4]. In the REACH registry [5], the subgroup of patients with both CAD and PVD had a higher incidence of cardiovascular events compared with those with CAD only. In a wade National US [6] insurance database of AMI patients, those with additional PAD had the worst outcome compared with those with additional CeVD or aortic disease.

PVD should therefore be considered a strong predictor of cardiovascular events and a potential marker of coronary artery disease. However, the relative prognostic weight of the involvement of peripheral and cerebrovascular beds, either alone or in combination, in patients with coronary artery disease remains unclear. Moreover, these patients are often identified late in the course of the disease and are often undertreated. In fact, both conditions (CeVD and PAD) can be asymptomatic for many years and can be difficult to diagnose at an early stage until an adverse cardiovascular event occurs.

In this study, we seek to investigate the prevalence of symptomatic PVD in a large, nationwide, administrative cohort of patients with a new episode of acute coronary syndrome (ACS) and the impact of the involvement of either single or multiple vascular beds on long-term outcomes.

## 2. Material and Methods

This was a retrospective cohort study enrolling all patients admitted for an episode of ACS to any of the public and private hospitals in Italy from 1 January 2017 to 31 December 2018. The Italian National Registry of Hospital Discharge Records (HDR), provided by the Italian Ministry of Health (MoH), and other administrative databases available through a collaboration with the Italian National Program for Outcome Evaluation (PNE-AGENAS), were the data sources for this study.

The Italian HDR database was used to identify the study population. All HDRs of patients aged 40–100 years, resident in Italy, with a primary diagnosis of AMI [International Classification of Disease, 9th Revision, Clinical Modification (ICD 9 CM) 410], or a primary diagnosis of other acute or subacute form of ischaemic disease (ICD 9 CM 411) or a secondary diagnosis of AMI or other acute or subacute form of ischaemic disease with any concomitant complication within the primary diagnosis (ICD-9-CM codes 410, 411, 413, 414, 426, 427, 428, 423.0, 429.5, 429.6, 429.71, 429.79, 429.81, 518.4, 18.81, 780.01, 780.2, 785.51, 799.1, 997.02, and 998.2) or any diagnosis of Angina Pectoris (ICD 9 CM 413) were selected (Outcomes evaluation National Program [PNE] Ed. 2024; available at https://pne.agenas.it/; access on: 1 November 2024).

To minimise the inclusion of probable false AMI cases and multiple admissions for the same event, patients discharged home within 2 days of admission with an AMI diagnosis, duplicate records, and records relating to patients transferred to another hospital were excluded; moreover, multiple admissions occurring during 30 days after the index admission were considered as a single AMI episode.

Patients meeting the cohort definition criteria were stratified according to the presence of peripheral arterial disease (PAD) only, cerebrovascular disease (CeVD) only, PAD and CeVD (PAD+CeVD), or neither (noPAD/noCeVD, i.e., only CAD) in the index admission or in any hospitalisation over the past 6 years. Patients with PAD, CeVD, or both (in addition to CAD) were considered as having PVD. Patients were then classified as PAD+CeVD if they had at least one admission with indication of PAD and one admission with indication of CeVD in the index or in any hospitalisation over the previous 6 years (PAD and CeVD could be present in the same admission or in different admissions); if neither of the two conditions was present in any admission, patients were classified as noPAD/noCeVD. ICD9-CM codes used to identify the exposure categories are reported in Appendix A [7].

The risk factors and comorbidities used to define the severity of the enrolled patients were identified by the ICD 9 CM codes recorded either in the index hospitalisation or in any hospitalisation in the previous 3 years (Appendix A).

### 2.1. Outcomes

Major adverse cardiac and cerebrovascular events (MACCE) (i.e., death from any cause, AMI, or ischemic stroke [IS]) was considered as the main adverse outcome. The incidence of all-cause mortality, new admissions for heart failure (HF), IS and AMI were also evaluated as secondary endpoints.

### 2.2. Statistical Analysis

Results were stratified by exposure categories (noPAD/noCeVD, PAD only, CeVD only, and PAD+CeVD). Univariate comparisons of baseline characteristics by exposure categories were tested using the ANOVA and the Chi-squared test. Unadjusted differences in the cumulative incidence were evaluated using the Kaplan–Meier method with the log-rank test for MACCE and all-cause death, whereas a competing risk approach with the Gray test (death as a competing event) was used for the other non-fatal outcomes. To provide adjusted data on time to MACCE and all-cause death, age, gender, and patients’ risk factors and comorbidities were included in a multivariate Cox proportional hazards model as potential confounding factors; stepwise procedures were used to identify independent associations with MACCE and with all-cause death. Proportional hazards were checked using Shoenfeld residuals and the Kolmogorov–Smirnov test. The competing risk approach with the Fine-Grays method was used for time-to-event adjusted analysis of non-fatal outcomes (death as a competing event). Risk estimates were reported as hazard ratios (HR) or sub-hazard ratios (SHR) with their 95% confidence interval (CI) and *p*-value. Statistical analyses were performed using SAS 9.4 (Cary, NC, USA).

## 3. Results

The study flow chart is shown in Figure 1. Baseline characteristics of the study population are shown in Table 1. Out of 342,052 ACS patients, 201,269 (58.8%) had a diagnosis of AMI in the index admission, 69,580 (20.3%) of angina pectoris, and 71,203 (20.8%) of other ACS. Out of these, 24,747 (7.2%) patients had PAD only, 16,887 (4.9%) had CeVD only, and 5810 (1.7%) had both PAD and CeVD. Patients with PVD were older and had a higher prevalence of previous chronic ACS, coronary revascularisation (PCI and/or CABG), heart failure, diabetes, and other chronic diseases compared with the reference group (294,608 patients with ACS but neither PAD nor CeVD) (Table 1).

Figure 2 shows the cumulative incidence estimated by the Kaplan–Meyer and the Fine competing risk survival analysis for the outcomes under consideration. Patients with PVD always had a worse 5-year survival than those with ACS alone. In particular, patients with PAD+CeVD had the highest 5-year incidence rate for each outcome analysed (except for readmission for stroke); among the remaining groups, patients with PAD only had a worse 5-year incidence rate for all-cause mortality and readmission for HF and AMI compared with those with CeVD only and those with ACS only. Patients with CeVD only had the worst 5-year incidence of readmission for stroke (Figure 2 and Appendix A). After adjusting for age, sex, and comorbidities, the hazard ratios (HRs) for 5-year MACCE in patients with PAD only, CeVD only, and PAD+CeVD were 1.37 (95% CI: 1.35–1.40), 1.36 (95% CI: 1.30–1.39) and 1.45 (95% CI: 1.40–1.50), respectively, compared to patients without PAD and CeVD (Table 2 and Figure 3).

Table 3 shows the adjusted 5-year HRs/SHRs for all-cause mortality, AMI, stroke and HF. The 5-years adjusted HRs for all-cause mortality in patients with PAD only, CeVD only, and PAD+CeVD were 1.48 (95% CI: 1.45–1.51), 1.32 (95% CI: 1.27–1.33), and 1.52 (95% CI: 1.46–1.57), respectively, compared with patients with ACS only (Table 3). Patients with CeVD showed a 5-year risk of readmission for stroke significantly higher than patients without CeVD, particularly when compared with NoPAD/NoCeVD patients (HR = 3.22; 95% CI: 2.90–3.57 for PAD+CeVD and HR = 3.84; 95% CI: 3.60–4.09 for patients with CeVD only). Conversely, patients with PAD had a higher 5-year risk of hospital readmission for AMI or HF than the other groups analysed, in particular when compared with patients with ACS only (AMI: HR = 1.27; 95% CI: 1.24–1.31 for PAD pts, and HR = 1.32; 95% CI: 1.25–1.40 for PAD+CeVD pts; HF: HR = 1.32; 95% CI: 1.29–1.36 for PAD only pts, and HR = 1.34; 95% CI: 1.27–1.41. for PAD + CeVD pts.) (Table 3). Appendix A shows the multivariate models used for each outcome.

## 4. Discussion

The main findings of this analysis of a nationwide, universal administrative database of consecutive adult ACS patients are the following: (1) ACS patients with the involvement of a second vascular bed showed an approximately 35% increased 5-year risk of MACCE compared with patients with ACS alone; (2) simultaneous involvement of three vascular beds further increases the risk of long-term outcomes (45% excess risk).

Atherosclerotic cardiovascular disease (ASCVD) is the most common cause of death globally [1]. As a systemic disease, it can affect one or more vascular beds (coronary, carotid/cerebrovascular and/or peripheral arteries) leading to coronary artery disease (CAD) or coronary syndromes (ACS), cerebrovascular disease (CeVD, i.e., ischemic stroke [IS] and or TIA) and peripheral artery disease (PAD). Based on the NHANES 2017–2020 data, the overall stroke prevalence in the United States (US) during this period was estimated to be 3.3% [1], with a 20.5% increase in stroke prevalence by 2030 (an additional 3.4 million US adults aged ≥18 years of age representing 3.9% of the adult population) [8]. According to the GBD 2019 study, ischemic strokes accounted for 62.4% of all global incident strokes (7.63 [95% CI, 6.57–8.96] million) worldwide in 2019 [9]. In the European Union, in 2017, there were 1.12 million incident strokes, 0.46 million deaths, and 7.06 million disability-adjusted life years lost due to stroke. It has been estimated that by 2047 there will be an additional 40,000 incident strokes (+3%) and 2.58 million prevalent cases (+27.0%) [10]. The global prevalence of symptomatic carotid stenosis was about 1.5%, and interestingly this increased to 8.50 million people aged over 65 years in 2020 (6.58–10.95) [2].

Peripheral artery disease (PAD) is a major cardiovascular disease that affects 202 million people worldwide. Overall, the estimated 2020 global prevalence of PAD in the general population was about 8.0% in people aged 65–69 years and increases exponentially with ageing (about 12% in people aged 70–79 years in both men and women in high-income countries) [3,11].

Patients with polyvascular disease (PVD) show a high risk of major adverse cardiovascular events (MACCE) such as myocardial infarction, stroke, and cardiac death [4]. Among patients with CAD, PAD is present in approximately 50% of cases, whereas carotid atherosclerosis is present in up to 20%. The involvement of multiple vascular beds is associated with a greater severity of the disease and a worse prognosis [12]. In the REACH registry (REduction of Atherothrombosis for Continued Health) [5], the subgroup of patients with both CAD and PVD (18% of the overall enrolled symptomatic population) had a higher incidence of cardiovascular events (myocardial infarction/stroke/rehospitalisation/death) and the highest event rate at 1- and 3-year follow-up. In the wade National US Inpatient Sample [6] insurance database, which included 2,184,614 patients admitted for AMI between 2015 and 2017, approximately 50% of them had a disease affecting at least one vascular district. Patients with PVD involving three vascular beds showed a higher incidence of both total mortality and ischemic stroke events. Among patients with CAD who had a disease affecting a single other vascular bed in addition to the coronary one, patients with PAD had the worst outcome compared with those with CeVD or aortic disease. In another study using the same database of 1,441,000 patients who were candidates for percutaneous coronary intervention, the prevalence of peripheral vascular involvement was 14.2%. Compared with patients without involvement of extracoronary vascular beds, patients with PAD had a total mortality rate approximately 20% higher than those without PAD. In summary, the prevalence of PAD in patients with CAD ranges from 14% to 50% [11,13,14] and the coexistence of the two conditions is associated with an increasing incidence of MACCE and worse mortality.

The prognostic burden of PVD has been recently confirmed by other observational and intervention studies. In the Ultimaster registry, which included 37,198 patients [15] who had undergone percutaneous coronary intervention (PCI), 5.1% had PVD. Patients were stratified based on the presence or absence of prior vascular disease involving a single or 2–3 vascular beds. The propensity score adjusted all-cause death at 1-year increased significantly with the number of diseased vascular beds (2.22, 3.28, and 5.29%). The GLOBAL LEADERS trial showed a significantly higher mortality in patients having PCI with established vascular disease in three territories compared with those without other vascular bed involvement [16]. Similarly, in the SYNTAXES study, the presence of CeVD involving more than one vascular bed was associated with a significantly increased risk of 10-year all-cause death [17].

In a single-centre retrospective longitudinal observational study including 1380 symptomatic PAD patients treated with statins, the association of PVD (i.e., PAD plus one additional vascular region, CAD or CeVD, or PAD plus two vascular regions, CAD and CeVD) with the risk of all-cause mortality was evaluated. PAD patients with PVD received better statin medication and reached the recommended LDL-C target compared with PAD-only patients (*p* < 0.001). Despite better statin treatment, all-cause mortality rate was higher among PVD patients than among PAD-only patients (PAD only: 13%; +1 vascular region: 22%; +2 vascular regions: 35%; *p* < 0.0001) [18].

PVD should therefore be considered a strong predictor of cardiovascular events and a potential marker of coronary artery disease. However, because the progression of atherosclerosis can be asymptomatic for many years, affected patients are often diagnosed late in the course of the disease and remain undertreated until an adverse cardiovascular event occurs. Therefore, the estimation of the epidemiological burden of PVD can serve as a basis for prevention and management of cardiovascular disease. Early diagnosis can allow prompt treatment with slowing of disease progression and consequent reduction in cardiovascular morbidity and mortality [19,20].

Two main pharmacological strategies have been shown to reduce events in patients with PVD: (1) intensive reduction in LDL cholesterol to a target level of <55 mg/dL or, in some cases, <40 mg/dL; (2) antithrombotic therapy with the combination of aspirin + low-dose anticoagulant (rivaroxaban). In a meta-analysis based on seven studies including 94,362 patients, (18.6% with PVD) in PVD patients, intensive lipid lowering treatment was associated with a significant 15% reduction in the primary end point (RR = 0.85; 95% CI: 0.80–0.91 [*p* < 0.0001]), absolute risk reduction: 6.5%, compared to less lipid lowering treatment [21].

A sub-analysis of the Odyssey Outcomes study (Evaluation of Cardiovascular Outcomes After an Acute Coronary Syndrome During Treatment with Alirocumab) investigated the efficacy of the PCSK9 inhibitor alirocumab in 18,924 patients, of whom 17,370 had monovascular (coronary) disease, 1405 had a PVD interesting two vascular beds (coronary and peripheral artery or cerebrovascular), and 149 had PVD in all three vascular beds (coronary, peripheral artery, cerebrovascular). Treatment with alirocumab led to an absolute risk reduction of 1.4% (95% CI: 0.6% to 2.3%), 1.9% (95% CI: 2.4% to 6.2%), and 13.0% (95% CI: 2.0% to 28.0%) in patients with one, two and three vascular beds involved, respectively, compared with placebo (incidence of MACCE of 10.0%, 22.2%, and 39.7%, respectively). Furthermore, patients with PVD in all three vascular beds showed a significant absolute risk reduction with alirocumab (16.2%; 95% CI: 5.5% to 26.8%) compared with standard therapy [22].

Similar results were found in a sub-analysis of the FOURIER (Further Cardiovascular Outcomes Research with PCSK9 Inhibition in Subjects with Elevated Risk) trial, which investigated the efficacy of PCSK9 inhibitor evolocumab on acute arterial events across all vascular territories, including coronary, cerebrovascular, and peripheral vascular beds, in patients with stable atherosclerotic cardiovascular disease (ASCVD). Out of 2210 first acute arterial events, 74% were coronary, 22% were cerebrovascular, and 4% were peripheral vascular. Evolocumab reduced first acute arterial events by 19% (HR = 0.81 [95% CI: 0.74–0.88]; *p* < 0.001), with significant reductions in acute coronary (HR = 0.83 [95% CI: 0.75–0.91]), cerebrovascular (HR 0.77 [95% CI: 0.65–0.92]), and peripheral vascular (HR 0.58 [95% CI: 0.38–0.88]) events. The magnitude of reduction in acute arterial events with evolocumab increased over time, with a 16% reduction in the first year and 24% thereafter [23].

The second main pharmacological strategy in patients with PVD, particularly in those with lower limb PAD, is the antithrombotic therapy with a combination of aspirin + low-dose rivaroxaban. This strategy is mainly based on data from the COMPASS study which enrolled 27,395 participants with stable atherosclerotic vascular disease. The study showed that patients treated with rivaroxaban (2.5 mg twice daily) plus aspirin had better cardiovascular outcomes compared with those treated with aspirin alone [24]. These results were confirmed in the sub-group of patients with coronary artery disease who had a myocardial infarction in the past 20 years. In this analysis, the combination of aspirin + low dose rivaroxaban was more effective than aspirin alone in reducing the primary outcome (4% of 8313 vs. 6% of 8261; HR = 0.74, 95% CI: 0.65–0.86, *p* < 0·0001), resulting in an efficacy of treatment even for a median of seven years from the first cardiovascular event [25]. In another sub-study from the COMPASS, 7470 patients with stable PAD or carotid artery disease were selected from 558 centres. The combination of rivaroxaban plus aspirin compared with aspirin alone reduced the composite endpoint of cardiovascular death, myocardial infarction, or stroke (5% of 2492 vs. 7% of 2504; HR = 0.72, 95% CI: 0.57–0.90, *p* = 0.0047), and major adverse limb events including major amputation (1% vs. 2%; HR = 0.54; 95% CI: 0.35–0.82, *p* = 0.0037) [26].

Therefore, the combination of aspirin and low-dose rivaroxaban has been recommended in the European guidelines for patients with coronary syndromes at high or moderate residual thrombotic risk, identified by the presence of polyvascular disease (CAD plus PAD) [27], and in a statement by the ESC working group on aorta and peripheral vascular diseases, the ESC working group on thrombosis, and the ESC working group on cardiovascular pharmacotherapy [28].

Cardiac rehabilitation is the third intervention strategy, in this case non-pharmacological, proven to be effective in patients with coronary syndrome and PVD [29,30]. It has been shown that patients with PVD benefit from multidisciplinary interventions of intensive secondary prevention including cardiac rehabilitation, both because of the proven efficacy of physical exercise on the pathophysiological progression of vascular disease, even in patients with PAD [31], and because of the improvement in therapeutic adherence and long-term prognosis.

In conclusion, given the significant impact of polyvascular disease on the long-term prognosis of patients with coronary artery disease, increasing knowledge of the prevalence of symptomatic polyvascular disease in such patients is of considerable importance in guiding intensive secondary prevention programmes, thereby reducing re-hospitalisation and increasing cost-efficacy of care in this type of patients [32].

## 5. Study Limitations

This study has several limitations. As reported in the Section 2, the diagnosis of peripheral arterial disease (PAD) and cerebrovascular disease (CeVD), and thus PVD, was derived from hospital discharge records using ICD-9 coding. The lack of specific clinical information and the unavailability of imaging confirmation may have affected the accuracy of the diagnosis. Therefore, the prevalence of PVD may be underestimated, especially in the case of subclinical forms that may still have a prognostic impact. Another limitation is that the ICD-9 CM codes could not provide sufficient information on some clinical variables important for defining patient risk profile. As a result, some residual confounding may still be present. In any case, the use of administrative databases represents a specific strength of this study, as it guarantees the completeness of both enrolment and follow-up in terms of survivals and re-hospitalisations.

## 6. Conclusions

In patients with ACS, the involvement of a second vascular bed increases the risk of long-term outcomes; the simultaneous involvement of three vascular beds further increases this risk. These patients should be accurately identified and referred to intensive and cost-effective secondary prevention and rehabilitation programmes.

## Figures and Tables

**Figure 1 jcm-14-04158-f001:**
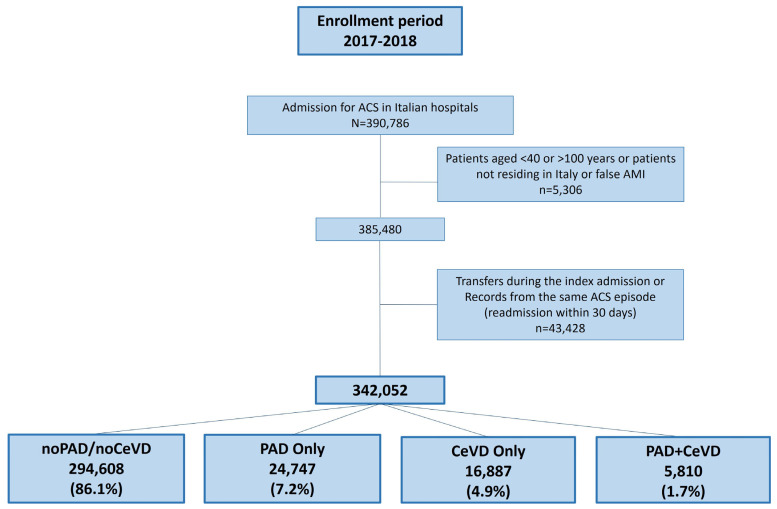
Study flow-chart. Selected acute coronary syndrome (ACS) patients were classified as follows: noPAD/noCeVD, PAD only, CeVD only, or PAD+CeVD. Total patients with PAD = 30,557 (8.9%), total patients with HF = 22,697 (6.6%).

**Figure 2 jcm-14-04158-f002:**
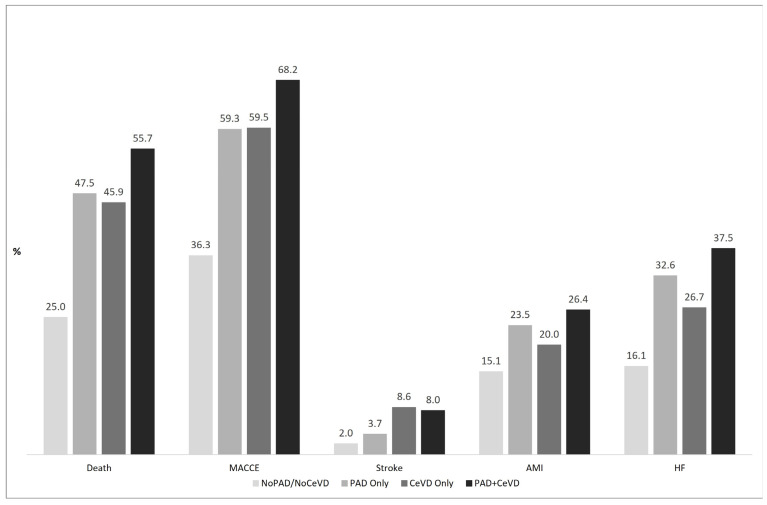
Five-year outcomes in different vascular disease combinations. Abbreviations: MACCE = Major Adverse Cardio-Cerebrovascular Events; AMI = Acute Myocardial Infarction; HF = Heart Failure.

**Figure 3 jcm-14-04158-f003:**
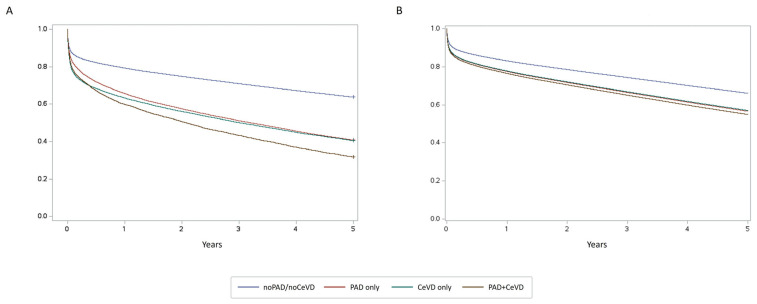
Kaplan–Meier survival curve (**A**) and Cox Adjusted survival curves (**B**) for 5-year MACCE. Violet line: noPAD/noCeVD patients. Green line: CeVD only patients. Red line: PAD only patients. Brown line: PAD+CeVD patients.

**Table 1 jcm-14-04158-t001:** Baseline characteristics of the enrolled population.

	PVD Categories					TOT (n = 342,052)
	noPAD/noCeVD (n = 294,608)	PAD Only (n = 24,747)	CeVD Only (n = 16,887)	PAD+CeVD (n = 5810)	*p*-Value	
Age	69.3 ± 12.3	72.5 ± 10.4	75.3 ± 10.4	74.5 ± 8.9	<0.0001	70 ± 12.1
Female gender	93,866 (31.9)	6909 (27.9)	6194 (36.7)	1655 (28.5)	<0.0001	108,624 (31.8)
Diabetes	21,211 (7.2)	7262 (29.3)	3291 (19.5)	2281 (39.3)	<0.0001	34,045 (10.0)
Obesity	3611 (1.2)	856 (3.5)	323 (1.9)	175 (3.0)	<0.0001	4965 (1.5)
Hypertension	39,895 (13.5)	8660 (35.0)	5996 (35.5)	2955 (50.9)	<0.0001	57,506 (16.8)
COPD	18,797 (6.4)	3549 (14.3)	2069 (12.3)	1133 (19.5)	<0.0001	25,548 (7.5)
Previous AMI	38,734 (13.1)	6446 (26.0)	3273 (19.4)	1751 (30.1)	<0.0001	50,204 (14.7)
Previous other chronic coronary syndromes	46,045 (15.6)	9446 (38.2)	4718 (27.9)	2709 (46.6)	<0.0001	62,918 (18.4)
Previous CABG	17,383 (5.9)	3438 (13.9)	1763 (10.4)	1004 (17.3)	<0.0001	23,588 (6.9)
Previous PCI	56,406 (19.1)	7747 (31.3)	3706 (21.9)	1916 (33.0)	<0.0001	69,775 (20.4)
Previous cardiac surgery other than CABG	4090 (1.4)	808 (3.3)	376 (2.2)	180 (3.1)	<0.0001	5454 (1.6)
Heart failure	48,350 (16.4)	7890 (31.9)	4597 (27.2)	2090 (36.0)	<0.0001	62,927 (18.4)
Rheumatic heart disease	4930 (1.7)	730 (2.9)	541 (3.2)	206 (3.5)	<0.0001	6407 (1.9)
Cardiomyopathy	6786 (2.3)	1180 (4.8)	578 (3.4)	309 (5.3)	<0.0001	8853 (2.6)
Arrhythmias	58,335 (19.8)	6598 (26.7)	5461 (32.3)	1848 (31.8)	<0.0001	72,242 (21.1)
Other chronic heart diseases	9812 (3.3)	1878 (7.6)	1068 (6.3)	506 (8.7)	<0.0001	13,264 (3.9)
Endocarditis and acute myocarditis	3857 (1.3)	704 (2.8)	329 (1.9)	153 (2.6)	<0.0001	5043 (1.5)
Anaemia	15,754 (5.3)	3397 (13.7)	1696 (10.0)	942 (16.2)	<0.0001	21,789 (6.4)
Malignant neoplasms	17,589 (6.0)	2202 (8.9)	1348 (8.0)	493 (8.5)	<0.0001	21,632 (6.3)
Chronic Kidney Diseases	26,083 (8.9)	6705 (27.1)	3013 (17.8)	1820 (31.3)	<0.0001	37,621 (11.0)
Coagulation disorders	350 (0.1)	57 (0.2)	49 (0.3)	16 (0.3)	<0.0001	472 (0.1)
Other chronic diseases (liver, pancreas, bowel)	385 (0.1)	72 (0.3)	46 (0.3)	24 (0.4)	<0.0001	527 (0.2)
Diagnosis in index admission						
AMI	172,658 (58.6)	14,286 (57.7)	10,774 (63.8)	3551 (61.1)	<0.0001	201,269 (58.8)
Angina pectoris	61,664 (20.9)	4416 (17.8)	2594 (15.4)	906 (15.6)		69,580 (20.3)
Other acute and subacute forms of ischemic heart disease	60,286 (20.5)	6045 (24.4)	3519 (20.8)	1353 (23.3)		71,203 (20.8)

Data are reported as mean + SD for continuous variables and as counts and percentages (in brackets) for dichotomous variables. Abbreviations: COPD = Chronic Obstructed Pulmonary Diseases; AMI = Acute Myocardial Infarction; CABG = Coronary Artery Bypass Graft; PCI = Percutaneous Coronary Intervention.

**Table 2 jcm-14-04158-t002:** Multivariate Cox regression model for 5-year MACCE. Hazard Ratio (HR), confidence interval (CI), *p*-value.

	HR	95% CI		*p*-Value
PAD only	1.375	1.351	1.400	<0.0001
CeVD only	1.357	1.329	1.386	<0.0001
PAD+CeVD	1.449	1.402	1.496	<0.0001
Female gender	0.952	0.941	0.963	<0.0001
Age	1.043	1.042	1.043	<0.0001
Malignant neoplasms	1.397	1.371	1.422	<0.0001
Diabetes	1.197	1.177	1.217	<0.0001
Obesity	1.107	1.063	1.153	<0.0001
Anaemia	1.274	1.251	1.296	<0.0001
Previous AMI	0.98	0.964	0.995	0.0118
Previous other chronic coronary syndromes	0.919	0.904	0.934	<0.0001
Heart failure	1.557	1.537	1.577	<0.0001
Rheumatic heart disease	1.153	1.118	1.189	<0.0001
Cardiomyopathy	1.11	1.079	1.141	<0.0001
Arrhythmias	1.255	1.24	1.27	<0.0001
COPD	1.2	1.179	1.22	<0.0001
Chronic Kidney Diseases	1.37	1.35	1.39	<0.0001
Other chronic diseases (liver, pancreas, bowel)	1.306	1.258	1.357	<0.0001
Previous PCI	1.343	1.322	1.364	<0.0001
Endocarditis and acute myocarditis	1.242	1.111	1.387	0.0001
Diagnosis in index admission: Other ischemic heart disease	Ref.			
Angina pectoris	1.542	1.509	1.575	<0.0001
AMI	2.753	2.702	2.804	<0.0001

Abbreviations: COPD = Chronic Obstructed Pulmonary Diseases; AMI = Acute Myocardial Infarction; PCI = Percutaneous Coronary Intervention.

**Table 3 jcm-14-04158-t003:** Risk-adjusted 5-year all cause death and re-hospitalisation for HF, Ischaemic Stroke and AMI in patients having PAD only, CeVD only, and PAD+CeVD (reference noPAD or CeVD). Hazard Ratio (HR)/Sub-Hazard Ratio (SHR); Confidence Interval (CI).

	HR/SHR	95% CI	
All-Cause Death			
PAD only	1.476	1.446	1.506
CeVD only	1.295	1.265	1.326
PAD+CeVD	1.518	1.464	1.573
Rehospitalization for HF			
PAD only	1.321	1.285	1.358
CeVD only	1.156	1.117	1.196
PAD+CeVD	1.336	1.27	1.405
Rehospitalization for Ischaemic Stroke			
PAD only	1.541	1.427	1.665
CeVD only	3.836	3.601	4.085
PAD+CeVD	3.217	2.897	3.574
Rehospitalization for AMI			
PAD only	1.274	1.237	1.312
CeVD only	1.184	1.141	1.227
PAD+CeVD	1.321	1.250	1.395

Note: Risk-adjustment model for each outcome considered was reported in Appendix A.

## Data Availability

The data underlying this article were made available by AGENAS and will be shared upon reasonable request to the corresponding author with the permission of AGENAS.

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
