# Peer review of "Impact of Polyvascular Disease on Long-Term Prognosis of Patients with Acute Coronary Syndrome—A Retrospective Cohort Study in Italy"

_jcm, 2025, doi:10.3390/jcm14124158_

Round 1

Reviewer 1 Report

Comments and Suggestions for Authors

I am grateful to the editor for the opportunity to review the manuscript of Gian Francesco Mureddu et al. "Impact of polyvascular disease on long-term prognosis of patients with acute coronary syndrome. A retrospective cohort study in Italy". This study is devoted to the problem of multifocal atherosclerosis, showed its prevalence in a national sample of patients with ACS in Italy over 2 years, as well as its impact on 5-year prognosis. No particularly new scientific facts were obtained; similar results were published earlier in other countries. This was shown both for hospital results in the CRUSADE registry (doi: 10.1093/eurheartj/ehp099) and during long-term follow-up (reference 12 in the article). Probably, such a study was important for Italian healthcare.
While reviewing, I had the following comments and questions:
1. The introduction does not sufficiently substantiate the novelty of this study. The authors provided references to studies with a similar design in other countries only in the Discussion section. Although it would have been appropriate to mention them in the Introduction section and justify the need for another study with a similar design.
2. In the Discussion section, the authors devoted much attention to the treatment of patients with multifocal atherosclerosis, although the treatment of this category of patients was not considered in the article. It would have been more appropriate to provide data on the prevalence of multifocal atherosclerosis and its impact on prognosis not only in patients with ACS, but also in other localizations of atherosclerosis (for example, ref. 1-2, see below). We can also recall the comment by Shigetaka Kageyama et al. on a similar matter (doi: 10.1093/ehjqcco/qcac048).
3. Although the article contains references to Suppl. Table 1a and Suppl. Table 1b, I was unable to find the data in the table in the presented materials. 4. Reference numbering using Latin numbers does not look good, I suggest using Arabic numerals after all.
References:
1. Adam L, Strickler E, Borozadi MK, Bein S, Bano A, Muka T, Drexel H, Dopheide JF. Prognostic Role of Polyvascular Involvement in Patients with Symptomatic Peripheral Artery Disease. J Clin Med. 2023 May 11;12(10):3410. doi: 10.3390/jcm12103410.
2. Kobo O, Saada M, von Birgelen C, Tonino PAL, Íñiguez-Romo A, Fröbert O, Halabi M, Oemrawsingh RM, Polad J, IJsselmuiden AJJ, Roffi M, Aminian A, Mamas MA, Roguin A. Impact of multisite artery disease on clinical outcomes after percutaneous coronary intervention: an analysis from the e-Ultimaster registry. Eur Heart J Qual Care Clin Outcomes. 2023 Jun 21;9(4):417-426. doi: 10.1093/ehjqcco/qcac043.

Author Response

1.The introduction does not sufficiently substantiate the novelty of this study. The authors provided references to studies with a similar design in other countries only in the Discussion section. Although it would have been appropriate to mention them in the Introduction section and justify the need for another study with a similar design.

Thank You very much for your comments and suggestions. The introduction section was changed accordingly. We have added references to studies with similar design and the following sentences (in red color in the revised text):

Line 50-54: “In the REACH registry (REduction of Atherothrombosis for Continued Health) the subgroup of patients with both CAD and PVD had a higher incidence of cardiovascular events compared with those with CAD only. In a wade National US insurance database of AMI patients, those wth additional PAD had the worst outcome, compared with those with additional CeVD or aortic disease”

Line 56-58: “However, the relative prognostic weight of the involvement of peripheral and cerebrovascular beds, either alone or in combination, in patients with coronary artery disease remains unclear”.

2.In the Discussion section, the authors devoted much attention to the treatment of patients with multifocal atherosclerosis, although the treatment of this category of patients was not considered in the article. It would have been more appropriate to provide data on the prevalence of multifocal atherosclerosis and its impact on prognosis not only in patients with ACS, but also in other localizations of atherosclerosis (for example, ref. 1-2, see below). We can also recall the comment by Shigetaka Kageyama et al. on a similar matter (doi: 10.1093/ehjqcco/qcac048).

Thank you for this significant observation which actually helps us improve our work. We have modified the discussion and cited some studies on the treatment of polyvascular disease (in particular we have cited, as you suggested, the work of Kobo et al., Garg et al., and Adam et al.) You can find the changes in the revised manuscript line 207-225:

“The prognostic burden of PVD has been recently confirmed by other observational and intervention studies. In the Ultimaster registry, which included 37,198 patients who had undergone percutaneous coronary intervention (PCI), 5.1% had PVD. Patients were stratified based on  the presence or absence of prior vascular disease involving a single or 2–3 vascular beds. The propensity score adjusted all-cause death at 1-year increased significantly with the number of diseased vascular beds (2.22, 3.28, and 5.29%). The GLOBAL LEADERS trial showed a significantly higher mortality in patients having PCI with established vascular disease in three territories compared with those without other vascular bed involved. Similarly, in the SYNTAXES study the presence of CeVD involving more than one vascular beds was associated with a significantly increased risk of 10-year all-cause death.

In a single-center retrospective longitudinal observational study including 1,380 symptomatic PAD patients treated with statins, the association of PVD (i.e. PAD plus one additional vascular region, CAD or CeVD, or PAD plus two vascular regions, CAD and CeVD) with the risk of all-cause mortality was evaluated. PAD patients with PVD received better statin medication and reached the recommended LDL-C target compared with PAD-only patients (p < 0.001). Despite better statin treatment, all-cause mortality rate was higher among PVD patients than among PAD-only patients (PAD only: 13%; +1 vascular region: 22%; +2 vascular regions: 35%; p < 0.0001)”(Adam et al).

With regard to this analysis, your suggestions are undoubtedly of interest. However, as stated in the introduction, the objective of this study is to focus on a cohort of patients with a new episode of acute coronary syndrome. Further studies on patients with atherosclerosis located elsewhere may be conducted in the future.

3.Although the article contains references to Suppl. Table 1a and Suppl. Table 1b, I was unable to find the data in the table in the presented materials.

We apologise for this inconvenience. We have resubmitted the supplemental material to the Editorial Office in the revised version of the paper.

4. Reference numbering using Latin numbers does not look good, I suggest using Arabic numerals after all.

In our original manuscript we used Arabic numerals. We apologise for this inconvenience. We have informed the Editorial Office accordingly.

Reviewer 2 Report

Comments and Suggestions for Authors

The authors provide a comprehensive retrospective cohort study using national administrative health data from Italy to assess the prevalence and prognostic impact of polyvascular disease (PVD) in patients hospitalized for acute coronary syndrome (ACS). The analysis includes over 340,000 patients and stratifies them based on the presence of peripheral artery disease (PAD), cerebrovascular disease (CeVD), both, or neither. The study convincingly demonstrates that involvement of additional vascular beds is associated with significantly worse long-term outcomes, including all-cause mortality and major adverse cardiac and cerebrovascular events (MACCE).

As they prepare a revised version of this review, I have several overall arching and some specific requirements and recommendations.  Individually and as a collective, these are meant to clarify the text, improve the Tables and provide further referencing and documentation in support of key background material.

First of all, as acknowledged by the authors, reliance on ICD-9-CM coding and administrative records may underestimate subclinical or underdiagnosed PAD and CeVD. This limitation should be more prominently discussed in the Abstract and Conclusion.

Secondly, the administrative nature of the data restricts access to clinically important variables such as LDL-c levels, medication adherence, smoking status, or imaging-confirmed diagnoses. The authors should more thoroughly address how this might bias results.

Moreover, the categorization of patients into “PAD only,” “CeVD only,” and “PAD+CeVD” based on any diagnosis in the index admission or previous 6 years may lead to misclassification due to changes in vascular status over time. Clarification on how repeat admissions were handled for exposure ascertainment would be helpful.

Finally, there are minor language issues persist (e.g., “sindrome” instead of “syndrome” in keywords; “hearth failure” instead of “heart failure”).

Author Response

The authors provide a comprehensive retrospective cohort study using national administrative health data from Italy to assess the prevalence and prognostic impact of polyvascular disease (PVD) in patients hospitalized for acute coronary syndrome (ACS). The analysis includes over 340,000 patients and stratifies them based on the presence of peripheral artery disease (PAD), cerebrovascular disease (CeVD), both, or neither. The study convincingly demonstrates that involvement of additional vascular beds is associated with significantly worse long-term outcomes, including all-cause mortality and major adverse cardiac and cerebrovascular events (MACCE).

As they prepare a revised version of this review, I have several overall arching and some specific requirements and recommendations.  Individually and as a collective, these are meant to clarify the text, improve the Tables and provide further referencing and documentation in support of key background material.

First of all, as acknowledged by the authors, reliance on ICD-9-CM coding and administrative records may underestimate subclinical or underdiagnosed PAD and CeVD. This limitation should be more prominently discussed in the Abstract and Conclusion.

Thank you for this suggestion, this limitation was given greater emphasis in the Study Limitations section. See the revised manuscript (in red color in the revised text) at lines 308-319.

Secondly, the administrative nature of the data restricts access to clinically important variables such as LDL-c levels, medication adherence, smoking status, or imaging-confirmed diagnoses. The authors should more thoroughly address how this might bias results.

Thank You very much for this meaningful comment. The section ‘Limitations of the study’ has been amended to address this remark. See the revised manuscript at lines 299-300 and lines 302-305.

Moreover, the categorization of patients into “PAD only,” “CeVD only,” and “PAD+CeVD” based on any diagnosis in the index admission or previous 6 years may lead to misclassification due to changes in vascular status over time. Clarification on how repeat admissions were handled for exposure ascertainment would be helpful.

Thank You very much for the opportunity to clarify the exposure definitions.

Repeated admissions and changes over time in patient vascular status were handled as follow:

  • "noPAD/noCeVD" refers to patients who don’t have any hospitalisation with an indication of PAD or CeVD, neither in the index nor in any previous 6-year hospitalisation;
  • "PAD only" refers to patients with at least one hospitalisation with an indication of PAD in the index or in the previous 6-year hospitalisation and no indication of CeVD in the index or in the previous 6 years hospitalisation;
  • "CeVD only" refers to patients with at least one hospitalisation with an indication of CeVD in the index or in the previous 6-year hospitalisation and no indication of PAD in the index or in the previous 6-years hospitalisation;
  • "PAD+CeVD' refers to patients with at least one admission with an indication of PAD and one admission with an indication of CeVD in the index or in the previous 6-year (PAD and CeVD may be present in the same or different admissions).

To better clarify this point, the following sentence has been added in the methods section:

“Patients were then classified as PAD+CeVD if they had at least one admission with indication of PAD and one admission with indication of CeVD in the index or in any hospitalisation over the previous 6 years (PAD and CeVD could be present in the same admission or in different admissions); if neither of the two conditions was present in any admission, patients were classified as noPAD/noCeVD”. See the revised text, lines 93-98.

Finally, there are minor language issues persist (e.g., “sindrome” instead of “syndrome” in keywords; “hearth failure” instead of “heart failure”).

We apologise for these mistakes. Requested corrections have been made and a thorough linguistic review has been carried out.

Round 2

Reviewer 1 Report

Comments and Suggestions for Authors

The authors responded to my comments and made corrections to the text. However, I still do not understand - why not use the numbering 1, 2, 3, etc. instead of i, ii, iii, etc.

Author Response

I completely agree to You. In the original manuscript they were Arabic. I have request to the Editorial Office to change the Latin in Arabic numbers. 

Reviewer 2 Report

Comments and Suggestions for Authors

The authors improved the manuscript. I have no others comments.

Author Response

Thank You very much for Your comment